# Entomologists in the K-12 Classroom: A Scoping Review

**DOI:** 10.3390/insects15100742

**Published:** 2024-09-26

**Authors:** Christopher B. Brown, Peter J. T. White

**Affiliations:** Department of Entomology, Michigan State University, East Lansing, MI 48823, USA; brow1249@msu.edu

**Keywords:** entomology education, STEM education, K-12, outreach

## Abstract

**Simple Summary:**

Entomologists, like many other types of scientists, will often share their scientific findings with the public. This is especially true for entomologists, who engage specific audiences, such as extension programs that target farmers, for example. But those focusing on K-12 classrooms are less well studied. Here, we conducted a scoping review to search for all relevant publications, and from those determine (1) the characteristics of K-12 entomology outreach efforts and (2) opportunities for improvement based on identified characteristics. The search of five databases yielded 42 relevant publications. Analysis of those publications identified the following characteristics of K-12 outreach efforts: These publications are most often published in educational journals, they are most often reflective, and they rarely evaluate the interventions employed. We suggest that the practice of K-12 entomology outreach benefits from (i) publishing in entomology-focused journals, (ii) including non-academic authors, (iii) evaluating interventions, (iv) including student data, and (v) considering elements of diversity and inclusion.

**Abstract:**

Engaging the public is a common practice in science disciplines and is deeply rooted in the discipline of entomology. These efforts to engage specific target groups within the general public are well studied, especially extension efforts to engage farmers and agricultural stakeholders, but this is not the case for K-12 educational spaces. Here, we conducted a scoping review to (1) determine the characteristics of entomology outreach efforts engaging K-12 populations and (2) identify opportunities for improvement based on the synthesis of those characteristics. We systematically searched five databases to identify 42 publications relevant to the parameters of this project. Analysis of characteristics indicated that entomology outreach efforts in K-12 classrooms tend to be reflective, are more often published in educationally focused journals, and rarely evaluate the interventions employed. Opportunities for improvement were identified from these trends, and from them we suggest that the practice of K-12 outreach benefits from (i) publishing in entomology-focused journals, (ii) including non-academic authors, (iii) evaluating interventions, (iv) including student data, and (v) considering axes of diversity and inclusion.

## 1. Introduction

Public outreach is an important component of scientific practice. However, academic research dissemination is most often achieved by publishing in scholarly journals or presenting at professional conferences. Both avenues are typically shielded from the general public through paywalls of journal subscription fees, or costly conference registrations. Occasionally, interesting tidbits of research are featured by news outlets or science podcasts, which, in some cases, are the only way research programs actually disseminate findings to the general public. This challenge has long been recognized, and in fact was one of the reasons why the National Science Foundation started to require that grant proposals include an explicit discussion of how each prospective project has a “broader impact” on the general public [1].

Some disciplines perhaps lend themselves to easier public dissemination and digestion than others. While the happenings in some research fields may seem somewhat obscure, impossibly complicated, and overly difficult to understand by a non-expert (for example, some academic branches of particle physics, mathematical number theory, or astrochemistry, to name just a few), other research fields, like entomology, can be much more relatable and have tangible connections to everyday life. For example, it is relatively simple to picture and conceptualize the general importance of insects as pollinators for both commercial crops and household gardens. Public awareness of pollinator importance is one of the central goals of non-profit groups like the Pollinator Partnership (pollinator.org) and the Bee Conservancy (thebeeconservancy.org). Conversely, the general public arguably has a basic understanding of the negative impacts of insects. Whether invasive crop pest species like the fall armyworm (*Spodoptera frugiperda*), household pests like the brown marmorated stink bug (*Halyomorpha halys*) or the Indian meal moth (*Plodia interpunctella*), or a forest pest like the emerald ash borer (*Agrilus planipennis*), insects constantly intersect with humans in significant, real, and sometimes personal ways. Given the relevance of insects to human society (in both positive and negative ways), the dissemination of entomological research findings to the general public should be a high priority. 

The public dissemination of entomology research findings is often targeted at one of three broad groups: (i) farmers, (ii) the insect-curious public, and (iii) children in K-12 education. Outreach and dissemination in agricultural spheres have long been recognized as part and parcel of agricultural entomology. The role of cooperative extension was created in 1914 with the Smith–Lever Act, specifically designed to *“[diffuse] among the people of the United States useful and practical information on subjects relating to agriculture*” [2], and today there are extension offices participating in outreach and public education in almost all 3000 US counties (nifa.usda.gov). Many of these programs operate in cooperation with local or regional commodity groups and focus on how to effectively manage pests while protecting pollinators to maximize crop yield. For example, researchers at Iowa State University used focus groups made up of farmers and agricultural stakeholders to develop an organic agricultural program [3]. 

Beyond partnerships with and dissemination to the agricultural industry, another popular target for entomology research outreach is what we have termed the “insect-curious public.” Many (if not most) entomology departments in colleges and universities throughout the US have some type of public outreach program, often characterized by fair-like events, “insect days”, or tourable insect museums, such as Insectapalooza at Cornell [4], Purdue’s annual Bug Bowl [5], and Michigan State University’s BugHouse [6]. These types of outreach programs and activities can have an impact on both the public perception and the public understanding of insects. Attendees to Virginia Tech’s Hokie BugFest were found to leave with improved attitudes towards insects [7], and educating participants was the sole purpose of 21% of the insect festivals reviewed in 2016 [8].

A third target for public outreach and academic research dissemination is students in the K-12 education system. Less scholarship exists focusing on this population compared to outreach efforts geared towards farmers and/or the insect-curious public. Scholarship occurring in K-12 spaces seems to be characterized in one of two ways: The first utilizes an entomology-based curriculum to investigate educational processes (e.g., Stroupe et al. [9], Grando et al. [10], Haefner et al. [11], etc.), and the second explores entomology-based concepts in the context of a K-12 classroom (e.g., Saunders et al. [12], Denton et al. [13], Braschler [14]). One inherent challenge in working in this amalgamated research space is that crossover entomology education researchers are rare; in these cases, efforts are made to incorporate best practices from both fields but often will not benefit from the involvement of a specialist in both entomology and education. Simply put, the field of “entomology education research” has never been formally defined or described. 

Our intention with this paper is thus to take some of the first steps needed to establish such a field. Here, we characterize and summarize entomological education outreach efforts involving K-12 populations in the form of a systematic scoping review. A “scoping review” is a type of review that aims to characterize and summarize an existing body of work on a given topic [15]. It is particularly useful to help establish, reshape, or refocus a scientific field (or sub-field). Specifically, we aim to determine (1) the characteristics of entomology outreach efforts engaging K-12 populations, and (2) the opportunities for improvement that can be identified from a synthesis of those efforts.

## 2. Methods

Our protocol followed the structures outlined in Arksey & O’Malley [16] and followed the PRISMA extension for scoping reviews.

### 2.1. Search Protocol

We used four major scholarly article database search tools to identify relevant entomological literature. Two database search tools were generally more oriented towards scientific and entomological literature (Web of Science and Agricola), including extension, and two database search tools were generally more oriented towards educational literature (Education Source and ERIC). Searches using these databases were conducted on 13 March 2023. In addition to these tools, we included the top 100 results in Google Scholar to account for literature that may have fallen between the gaps in our search process, which was conducted on 12 July 2023. We developed and used a search string designed to identify research that (1) included participants from educational settings, (2) included participants from scientific settings, (3) occurred in an educational setting, (4) was entomological in nature, and (5) included elements of outreach or engagements, or was lesson-based. We generated a set of words in each of these five categories (Table 1) and used them to produce a Boolean search string following the methodology established by Solano et al. [17]. An asterisk was included as a wildcard operator. Search terms within a category were separated by “OR”; search terms between were separated by “AND”. Google Scholar is unable to search using wildcard operators, so each relevant form of the identified words was manually entered into the search string. Scholarly articles identified using these methods were exported and compiled using Zotero; RIS files were then imported into Covidence for further categorization and processing.

### 2.2. Article Selection

Once the search process was completed, duplicate articles were automatically delimited in Covidence, and the titles and abstracts of each article were manually screened for relevance (Figure 1). Articles deemed “possibly relevant” at this stage were read in their entirety. It is the intent of this scoping review to characterize outreach efforts carried out by entomologists, engaging with K-12 students, via the use of entomology-focused content. This goal informed the creation of inclusion criteria that justified the inclusion of a publication if it met three requirements: (1) if the activities described in the paper occurred within an established formal education setting, (2) if the content used in the engagement directly relates to the biology or ecology of insects, and (3) if a member of academic science (e.g., an entomologist or a scientist in a related field) and a member of the public education system (e.g., one or more teachers and/or students) were included.

### 2.3. Article Coding

All papers that were identified in our article selection process were further coded using nine different characteristics. A custom-built data extraction template was created in Covidence to capture the predetermined qualities of each characteristic, with an option for “Other” that provided for a custom response to be entered. The coding was independently coded by a single author. 

Field of Publication: Articles could be published in (i) a primarily entomological journal, (ii) a scientific journal that is not primarily entomological, and (iii) a primarily educational journal.Type of Literature: Articles could be classified as (i) a journal article, (ii) a lesson plan, or (iii) a conference proceeding.Author Affiliation: Authors could be affiliated with (i) a higher-education institution, (ii) a government entity, (iii) a K-12 school or district, and (iv) no identifiable organization (i.e., an “independent” researcher or profession outside of education or science).Location: The geographic location (country and continent) where the research conducted in each article was classified for each paper.Types of Information Presented: Here, we looked at the types of information that each paper was presenting to support the findings or recommendations made. Types of information could include (i) a recounting of an outreach event, (ii) lesson materials and suggested implementation plans, (iii) data measuring student experience, (iv) ecological data about insects, (v) data measuring teacher experience, (vi) data measuring researcher experience, and (vii) a critical review of the established curriculum.Types of Participants: Our search terms allowed for articles to be included if there were representatives from a wide range of participants from both education and academic research environments involved. Here, we coded each article as including (i) teachers, (ii) students, (iii) entomological research scientists, and (iv) non-entomological researchers.Grades Engaged: Each paper was analyzed and categorized based on the grade level or grade-level band that the outreach was meant to engage.Intervention Type: Papers were categorized based on whether they used one type of intervention or more, and what the nature of the interaction was.The Inclusion of Student-Generated Data: Papers were categorized on whether the published work included ecological data that were student-generated, or whether the data presented were primarily (or solely) researcher-generated data.

Following data extraction, the results were exported from Covidence as a CSV file, and summary statistics were compiled for each characteristic. 

## 3. Results

### 3.1. Article Selection

The search parameters that we executed in five different database search tools yielded 1063 articles; once duplicate articles were removed, this number dropped to 961. After the articles were screened for relevance by reading the titles and abstracts, 121 remained. Full-text reading of these articles and applying our final selection criteria winnowed our set of articles down to 42.

### 3.2. Article Coding

Described below is an overview of the results from the data extraction of the included literature. A comprehensive display of relevant data coded from each source of evidence can be found in Appendix A. 

Field of Publication: Twenty-one of the final 42 articles were published in educational journals, 12 were published in entomological journals, 8 were published in journals of other scientific fields, and 1 was published in a journal that did not fit into any of these categories [18].Type of Literature: The majority of the final literature was published as journal articles (27), 12 as lesson plans, 2 as conference proceedings [10,19] and 1 as a book chapter [20].Author Affiliation: The greatest number of papers were authored by an academic professional with some higher education affiliation (37), nine were authored by governmental professionals, eight by public educators, and five by independent researchers or other career professionals.Location: The final 42 articles included research that spanned 5 continents and 13 countries. Thirty-one of the included documents concerned studies that took place in the Americas (30 in the US and 1 in Canada), five in Europe (2 in Germany and 1 each in England, Norway, Finland, and Italy), three in Africa (2 in South Africa and 1 in Ethiopia), three in Australia, two in the Middle East (both in Pakistan), and two in Asia (1 each in Japan and Russia). It was possible for publications to review projects that spanned multiple localities—for example, those that allowed entomologists to remotely engage with students in Kansas and Pakistan via Skype [18].Types of Information Presented: The majority of the papers presented a recounting of an outreach event (25) or provided structure and tips for conducting a similar outreach effort (15). It was possible for an article to both recount the experiences of the authors and suggest changes to be made in a proposed implementation of the same lesson. Other documents focused on a qualitative examination of the experiences of participating subpopulations, with ten of these focusing on those of students, four on those of teachers, and two on those of researchers. Only one article examined the experiences of both students and teachers [21], and one considered all three subpopulations [22]. Eight documents presented data collected by students in their experimental processes, and two were analyses of developed curricula [23,24]. These codes were not mutually exclusive; it was possible for any given document to recount the experiences of researchers as they conducted their outreach program, provide suggestions for future implementation, and provide the results of their analysis of participant experiences.Types of Participants: Our search terms allowed for articles with a wide range of participants from both education and academic research environments to be included. Here, we coded each article as including (i) teachers, (ii) students, and (iii) research scientists. (i) While teachers are assumed to have played some role in all of these activities, 28 of the 42 articles specifically mentioned teachers, their role, or their experiences. (ii) Similarly, 27 articles explicitly discussed student involvement. (iii) Entomological researchers were most often included in these discussions, appearing in 33 of the included articles, and other scientists were included least, in 19.Grades Engaged: Thirty-seven articles mentioned a specific grade level; five documents did not (Table 2). If an age range was reported in a non-US country, the equivalent grade was determined using the formula that generally works in the school systems of the US (Age 5 = grade level). While this scoping review was specifically meant to focus on K-12 groups, five papers additionally included engagement in post-secondary education settings. Authors that made mention of engagement involving middle-school children were assumed to refer to grades 6–8 and high-school children to grades 9–12.

8.Intervention Type: Thirty-one articles mentioned more than one type of intervention; only 11 mentioned a single intervention strategy. Data collection where students were positioned as technicians of a pre-designed experiment or were conducting and collecting data of their own experiments were the most common, present in 30 documents. Live insects were used in 28, and oration as a lecture or discussion in 25. Six took advantage of interactive digital media, with examples being remote control of an electron microscope [25] and virtual field trips [26]. Three documents discussed the use of videos as an intervention, and two did not mention interventions used with students [24,27].9.The Inclusion of Student-Generated Data: Student data were not represented in 34 of the included documents, and were present in 8.

## 4. Discussion

### 4.1. Characteristics of Entomology Outreach Efforts That Engage K-12 Populations

While publication-level scholarship that highlights the general spirit of outreach activities is beneficial for promoting and legitimizing those efforts, it does little to advance the practice of engaging K-12 populations. Of the 42 final publications, 22 of these presented neither ecological data nor data on participant experience, and in more than half of the cases (25), a recounting of events was the lone material presented. Inarguably, participants learned or viewed insects differently because of their experiences, but these effects were not often captured, as only 24% of these publications took into account the experience of students. Similarly, both the role and the experiences of the educator were often overlooked in these publications. In only four examples is teacher experience investigated [11,21,22,28], and while the teacher’s role was explained in many of the final publications (27), teachers were only represented as authors in eight. So, too, are changes in researcher affect rarely measured [18,22] as a result of these outreach activities. Admittedly, the results of the qualitative coding characterized these publications as mostly reflective and surface-level, focusing on the overarching narrative of an activity from the perspective of the researcher rather than providing metrics that can be used to make improvements to the practice or taking into account the experiences or perspectives of the participants.

Establishing and conducting a field for research to analyze the efforts of entomology outreach is going to require that publications be purposefully written for implicit use by other practitioners, but trends in the current literature indicate this practice is not widely adopted. In only 36% of the included literature are any materials or plans provided, or suggestions to be used in similar collaborations made. Only 2 of the final 42 publications expanded on or critiqued previously published outreach events [23,24]. This may suggest a field-wide tendency to treat curriculum of this type as single-use. Evaluation of interventions is a conserved best practice in the field of education research but is underrepresented in the outreach efforts included here; 69% included no measure of student, teacher, or researcher experience. 

One of the major benefits of engaging the general public via formal education spaces is that it can serve to forge connections with demographics otherwise not exposed to a professional level of science. Outreach efforts like these, which are taken into the community rather than hosted on a campus, have the potential to reach a more diverse audience and have a greater potential to elicit changes in participants [29]. Of the outreach efforts included in this scoping review, changes in the attitudes and perceptions of teachers and students were rarely measured (in only 4 and 10 cases, respectively). Outreach programs that ignore changes in participant attitudes and perceptions fail to capitalize on the positive products of public outreach that are assumedly taking place.

### 4.2. Opportunities for Improvement

Given our findings, we have identified five avenues for the improvement and advancement of K-12 entomology outreach. These opportunities are neither exhaustive nor prescriptive, but represent actionable steps to benefit the practice of effective K-12 entomology outreach. 

Publish in entomology-focused journals

Knight and Steinbach [30] suggest that the likelihood of manuscript publication is the single greatest factor in the journal selection process. This selection provides insight into both the intended audience of a manuscript and the assumed stakeholders of the information provided. Although all included literature was developed in some part by an entomologist, half (51%) of it was published in an educational journal. However, in order for the field of entomology education to be advanced in the entomological academic community, this type of research needs to be more commonplace in traditional entomology journals. This itself can be a limitation, as research papers that focus too much on education, in our experience, have a hard time getting into entomology-focused journals. This challenge can motivate entomologists who work in education spaces to publish on their work in alternative outlets, like educational journals. Whether or not this is an indication of selection bias on behalf of entomological journals, these publication patterns seem to indicate that those entomologists working in K-12 spaces likely will not get as much exposure to and recognition from their peers in traditional entomology research fields. To expose a greater number of entomologists to the educational efforts of their peers, publication in journals with specific entomological audiences is necessary. 

2.Include non-academic authors

Engagement of students in a formal K-12 setting is best carried out as a collaborative effort between researcher and educators, but these collaborative relationships are often not reflected in the authorship of the resulting publications. Classroom educators were noted as active participants in 66% of the included literature, and data were explicitly collected to describe their experiences in 9.8% of cases, but educators were only represented as authors in 19.5% of publications. These activities, and the publications that resulted from them, were only possible due to the willingness, flexibility, and structures provided by educators in each of these situations. In academia, authorship is an entrenched currency used in promotional processes and to determine merit. While these values are not shared by public educators, it cannot be assumed that they hold zero value. In the US, most public educators are compensated based on education and experience rather than performance [31], suggesting that the motivation to participate is intrinsic rather than financial or for professional gain. The contentious issues surrounding scholarly authorship are linked to diversity and inclusion and tend to disfavor minority groups [32], but authorship is ultimately a denotation of credit, and as such should include the efforts of all involved. An argument could be made, for example, that the activities of the collaborating schoolteacher do not meet the standards required for authorship, as typically outlined by journal guidelines. In some situations, this may certainly be the case. However, in other cases, the development and implementation of K-12 classroom activities relies heavily on the support and expertise of the associated classroom teachers. While it is not unreasonable to think that these somewhat uncommon types of contributions do not merit authorship, those operating in the entomology education research landscapes could be encouraged to think more creatively and broadly about the threshold at which teacher contributions merit co-authorship. By including non-academic collaborators, not only is credit being given for their effort but also the realm of academic science is shared with the public and a step is taken to increase representation in the field. 

3.Evaluate your interventions

Evaluation is an integral step in the practices of teaching and learning [33]. Evaluations can reflect the experiential or academic learning of students, but without evaluation the effectiveness of an educational activity cannot be assumed. A total of 76% of the included literature lacked any data collection or evaluation of student experience or assessment of student learning. It must be assumed by these numbers that in 76% of cases, entomologists are failing to evaluate the effectiveness of their designed materials and will be unable to attest to the efficacy of the created curriculum. This lack of evaluation may indicate a disposable mentality of entomology curricula: that an activity is single-use, designed for a specific classroom or teacher and then not used again. More efficient crafting of curricula involves designing an activity that can be used again, shared, and borrowed between classes and teachers, and distributed to a wider audience. To legitimize the time and energy of implementing a designed curriculum, teachers—with their limited planning time, resources, and often set curricular pacing—will require some degree of trust. In having evaluated a single activity or a set of activities, designers are able to pitch their activity with some confidence and have a product to deliver that has been subjected to revisions. 

4.Include student data

In our scoping review, we found that entomologists can engage students in K-12 spaces for a variety of reasons. However, the most common reason for engagement was to involve students in civic science types of projects. These projects put students in a position to answer real-world problems using student-collected data. Use of this practice engages students in a more authentic STEM experience and is supported by the Next Generation Science Standards [34], not only to develop student understanding of science core competencies but also to expose them to the nature and practice of science. In fact, a vast majority of studies (i.e., 30 of 42) involved civic science types of projects where students collected authentic ecological data. Of these 30, only 8 presented actual representations of that student-generated data (i.e., whether as summary statistics or as embedded in the study conclusions). For the remaining 22, the omission of student-collected data could run the risk of delegitimizing student work and misses an important opportunity for students to feel represented within the larger scientific community. 

5.Consider axes of diversity and inclusion

Effective entomology education outreach in K-12 settings has the potential to expand the inclusion of underrepresented minoritized groups in the STEM workforce [35]. There is a well-documented disparity in terms of representation and participation across various cultural and ethnic groups in America. As of 2020, minoritized groups (e.g., Hispanic Americans, African Americans, and American Indians or Alaska Native Americans) accounted for up to 27% of the US working-age population. Yet, they represented only 11% of the STEM workforce [36]. This lack of diversity in STEM fields is attributed, at least in part, to issues concerning exposure and representation [37]. In general, when young people are unaware of STEM career options and are unable to see themselves reflected in STEM professionals, they are less likely to choose STEM-based courses or pursue STEM-based degree programs [38]. Students engaged in the included outreach activities took on the role of scientific technicians in 71% of cases, yet student-generated data were included in only 19%. Herein lies an opportunity for researchers to adopt inclusionary practices and facilitate representation within the field. Researchers entering K-12 spaces can consider these axes of diversity and inclusion in a variety of other ways. Another option is for researchers to report student demographics in their publications. Taking care to follow federal privacy policies, researchers can report generalized student demographic information (e.g., Gall et al. [29]) that not only can inform the effectiveness of the materials being presented but also can be used in multi-study reviews given enough data. Currently, very little of the present literature discusses student demographics, and a comprehensive and informative analysis examining student demographics across multiple outreach events is not possible. All of these outreach events are making positive progress to engage students and hopefully foster a more diverse future for the STEM field, but by making small changes to the practices used in research design, the effects of these efforts and their effectiveness can be improved.

## 5. Limitations

During the process of applying the exclusion and inclusion criteria, a major limitation was discovered in the literature surrounding entomology education. A total of 17 published articles were found that fit all inclusion criteria save for participation by a scientific researcher. This study could have included 41% more literature to be analyzed had an entomologist (or adjacent scientist) been involved in those activities, many of which implemented live insect experiences and positioned students as scientists to develop and test their own hypotheses. This suggests that there are untapped opportunities for entomologists to collaborate with public educators and enrich an established entomological education program.

## 6. Conclusions

Intentional partnerships between researchers and teachers can play an enhanced role in field-wide efforts to increase diversity and inclusion in the STEM field. Greater effort to formalize and advance this sub-field of entomological education can be a positive step towards this goal. To strengthen the efficacy and effectiveness of entomological outreach and education, it is essential to address gaps and deepen understanding of the field. This scoping review sought to characterize entomology outreach and education efforts engaging K-12 populations and provide suggestions for improvement. Future directions for advancing this subfield could involve concerted efforts to publish entomology education research in entomology journals, engage in discussions surrounding inclusive authorship practices, evaluate classroom interventions, include student data in research publications (where appropriate), and consider how education outreach intersects with axes diversity and inclusion. 

## Figures and Tables

**Figure 1 insects-15-00742-f001:**
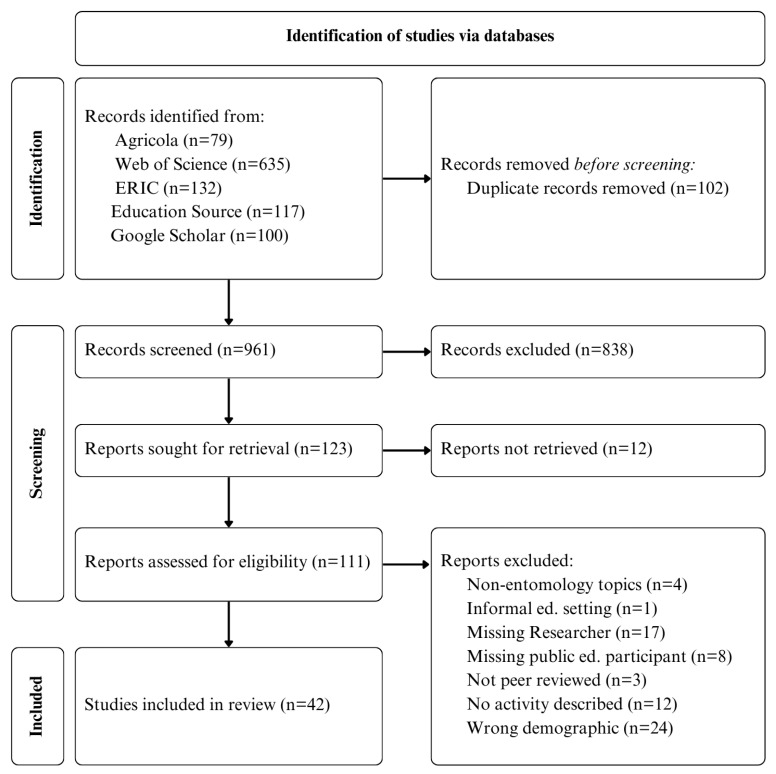
PRISMA flow chart of study selection process.

**Table 1 insects-15-00742-t001:** Categories and search terms used to create a Boolean search string to identify literature on entomologists in public education settings in electronic databases.

**Category**	**Educational Participants**	**Scientific Participants**	**Setting**	**Topic**	**Activity**
Search Term	student * teacher * educat * grade *	research * scientist *	school * class * grade * educat *	insect * entomolog * arthropod * bug *	outreach * engag * lesson *

Asterisk (*) used as a wildcard operator to include all declensions (e.g., the term “engag *” includes words such as engage, engages, engaging, etc.).

**Table 2 insects-15-00742-t002:** The number of publications engaging students within each grade band.

Grade Band Engaged	Number of Publications
K-5	9
K-5 & 6–8	5
K-5 & 6–8 & 9–12	11
6–8	2
6–8 & 9–12	4
9–12	6
Unspecified	5

## Data Availability

No new data were created or analyzed in this study. Data sharing is not applicable to this article.

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
