# Peer review of "Entomologists in the K-12 Classroom: A Scoping Review"

_insects, 2024, doi:10.3390/insects15100742_

Round 1
Reviewer 1 Report
Comments and Suggestions for Authors
The manuscript, "Entomologists in the K-12 classroom: A scoping review" provides information that would be valuable to entomologists engaging in public education programs, particularly K-12 educational spaces. The manuscript identifies a number of useful scholarly journal articles that discuss entomology programs in public education settings. The authors also provide constructive feedback for improving entomology outreach efforts. Overall, the manuscript is well thought out and written. However, there are several areas that would benefit from further discussion/clarification (see below comments).
Line 79: use lowercase e in “Entomology”.
Lines 89-90: please provide a citation. How is this area less well defined than other entomology outreach efforts that target the public/farmers?
Lines 114-117: Were Extension-focused journals captured in these searches? It would be a good idea to mention this somewhere in this section.
Line 197: Please revise this sentence.
Line 244: Where do these numbers come from? Based on Table 2, shouldn’t there be 22, 22, and 25 publications involved in engaging grades 6, 7, and 5, respectively?
Lines 280-282: This is an opinionated statement that should be revised or cited. I agree that in some cases publications are used to flaunt a completed outreach event. However, I do not believe that a lack of criticism in a manuscript generally means this is the case. For example, the purpose of some manuscripts are to spread excitement within the entomology community, document what others are doing, or showcase ways to target a specific audience.
Lines 302-317: Here the authors discuss the lack of publications in entomology focused journals. In lines 309-312, they mention this trend may result as a “product of hardship experienced by entomologists seeking to publish” these efforts. What are these hardships? In my experience, the biggest limitation to publishing K-12 studies in entomology journals is finding a journal that will accept these types of manuscripts. In Appendix A, the bulk of entomology journal articles are coming from only two journals – Insects and American Entomologist. Please discuss these limitations/hardships in more detail.
Lines 321-324: Here the authors discuss the lack of classroom educators (i.e. non-academic authors) being included as authors. One reason for this that was not discussed is the authorship contribution guidelines provided by journals. In many cases, authors must provide substantial contributions to the conception or design of the work, or the acquisition, analysis, or interpretation of data for the work. Please discuss in more detail how to include classroom educators in these processes.
Lines 345-347: This is an opinionated statement that should be revised or cited. I agree with everything the authors say before and after this sentence. However, there may be a number of reasons for why an evaluation is lacking that do not involve a “disposable mentality”. Constraints on time and effort are two that come readily to mind.
Line 418: What exactly was PW’s author contributions besides editing?
Figure 1: In the “reports excluded” box the authors mention that 8 reports were excluded because they were missing students. However, in line 147 they mention that papers were included if they had “one or more teachers and/or students”. The 8 excluded papers were missing students, but could they have involved teachers?
Comments on the Quality of English LanguageThe manuscript is well written and only requires minor editing (see comments for specific suggestions).
Reviewer 2 Report
Comments and Suggestions for Authors
Please see the attached comments and edits

A few issues with tense and active voice that is specified in the attached document.
Round 2
Reviewer 2 Report
Comments and Suggestions for Authors
The revision is okay for the most part. I have found the following that still needs editing for clarity:
Line 114-115 - current: "Both science focused databases include journals catering to the practice of extension, and would yield results
from these types of journals if applicable."
Suggested revision: The sentence could be removed without loss of clarity. As written, it is confusing, especially the end clause following "extension".
Line 260 - current: "...presented neither ecological or data on participant experience..."
Suggested revision: ecological is an adjective without a noun. Should it read ecological data or data on participant experience?
Line 280 - current: "...field wide tendancy..."
Suggested revision: hyphenate field-wide
Line 304 - current: "entomology-education"
Suggested revision: do not hyphenate - entomology education
Lines 306-308 - current: "This itself can be a limitation, as research papers that are deemed focus too much on education, in our experience, can have a hard time getting into entomology-focused journals."
suggested revision: is focus supposed to be past tense? focused? If so, then remove "that are deemed" to help clarify the sentence. Also remove "can".
Line 309 - current: "...spaces to publish on their work alternative outlets, like educational journals."
Suggested revision: missing "in" (after work).
Line 312: remove hyphens.
Line 340: remove hyphen from teacher contributions
Line 366: maintain singular consistency be changing experiences to experience.
Comments on the Quality of English Language
Moderate editing of English language required.
